# Real-Time Monitoring of Blood Parameters in the Intensive Care Unit: State-of-the-Art and Perspectives

**DOI:** 10.3390/jcm11092408

**Published:** 2022-04-25

**Authors:** Rebecca Bockholt, Shaleen Paschke, Lars Heubner, Bergoi Ibarlucea, Alexander Laupp, Željko Janićijević, Stephanie Klinghammer, Sascha Balakin, Manfred F. Maitz, Carsten Werner, Gianaurelio Cuniberti, Larysa Baraban, Peter Markus Spieth

**Affiliations:** 1Department of Anesthesiology and Critical Care Medicine, University Hospital Carl Gustav Carus, 01309 Dresden, Germany; rebecca.bockholt@ukdd.de (R.B.); shaleen.paschke@tu-dresden.de (S.P.); lars.heubner@ukdd.de (L.H.); alexander.laupp@tu-dresden.de (A.L.); 2Else Kröner-Fresenius Center for Digital Health (EKFZ), Technische Universität Dresden (TU Dresden), 01309 Dresden, Germany; bergoi.ibarlucea@tu-dresden.de (B.I.); z.janicijevic@hzdr.de (Ž.J.); sascha.balakin@tu-dresden.de (S.B.); gianaurelio.cuniberti@tu-dresden.de (G.C.); l.baraban@hzdr.de (L.B.); 3Institute for Materials Science and Max Bergmann Center of Biomaterials, Center for Advancing Electronics Dresden, Technische Universität Dresden, 01069 Dresden, Germany; stephanie.klinghammer@tu-dresden.de; 4Institute of Radiopharmaceutical Cancer Research, Helmholtz Center Dresden Rossendorf e.V., Bautzner Landstrasse 400, 01328 Dresden, Germany; 5Leibniz-Institut für Polymerforschung Dresden e.V., 01069 Dresden, Germany; maitz@ipfdd.de (M.F.M.); werner@ipfdd.de (C.W.)

**Keywords:** monitoring, intensive care, critical care, blood parameters, electrochemistry

## Abstract

The number of patients in intensive care units has increased over the past years. Critically ill patients are treated with a real time support of the instruments that offer monitoring of relevant blood parameters. These parameters include blood gases, lactate, and glucose, as well as pH and temperature. Considering the COVID-19 pandemic, continuous management of dynamic deteriorating parameters in patients is more relevant than ever before. This narrative review aims to summarize the currently available literature regarding real-time monitoring of blood parameters in intensive care. Both, invasive and non-invasive methods are described in detail and discussed in terms of general advantages and disadvantages particularly in context of their use in different medical fields but especially in critical care. The objective is to explicate both, well-known and frequently used as well as relatively unknown devices. Furtehrmore, potential future direction in research and development of realtime sensor systems are discussed. Therefore, the discussion section provides a brief description of current developments in biosensing with special emphasis on their technical implementation. In connection with these developments, the authors focus on different electrochemical approaches to invasive and non-invasive measurements in vivo.

## 1. Introduction

The number of patients treated in intensive care units (ICUs) is strongly increasing during the last decade. In the period between 1991 and 2018, the number of available ICU beds raised over 36% in Germany [1] and continues to grow. This highlights the more complex treatment, including various organ replacement procedures, e.g., renal replacement therapy or extracorporeal membrane oxygenation (ECMO). The main reasons for ICU admission involve sepsis, shock, postoperative care following complex surgery, trauma, myocardial infarction and acute respiratory distress syndrome (ARDS) [2]. ARDS is a common syndrome in critically ill patients, affecting approximately 10.4% of ICU patients and 23.4% of those mechanically ventilated [3]. In particular, the current SARS-CoV-2 pandemic highlights the importance of optimal ICU treatment and ECMO support in patients with severe ARDS [4,5]. The patient outcome mainly depends on the early recognition of evolving problems by the physicians in the ICU. Intensive patient monitoring is mandatory due to the high complexity of ICU treatment, where critically ill patients present rapid deterioration of vital signs. While cardiovascular and respiratory monitoring are usually available at the bedside in realtime using standard medical systems for hemodynamics and mechanical ventilation, laboratory blood parameter results are often delayed by at least several hours. Although disposable point-of-care devices are readily available at the bedside for specific parameters, e.g., blood gases and some coagulation tests, a substantial quantity of blood is required to apply those tests frequently. Studies have shown that the amount of blood used for testing is up to 40 mL during 24 h of stay in the ICU [6]. Given an average ICU stay of 10 days [3], testing results in approximately 400 mL of blood loss for laboratory analysis only. 

Therefore, bedside measurement of key blood parameters such as blood gases would certainly improve the quality of care towards more individualized therapeutic approaches. Moreover, the access to real-time blood lactate levels would allow for adequate treatment of critically ill patients [7,8]. Next to the capillary refill time [9], lactate is one of the most established markers indicating dysregulation of tissue (micro-) perfusion, and is frequently used in different states of shock, especially for guiding fluid resuscitation and differentiated catecholaminergic support [10,11]. In this review, different devices for real-time blood parameter monitoring are described and their effects on clinical applications is critically discussed. 

## 2. Aim and Method of This Review

Several devices are used for blood parameter measurements with different non-invasive and invasive methods. This narrative review aims to analyze and discuss the current literature regarding techniques for real-time blood parameter monitoring and their impact on intensive care and anesthesiology. Additionally, the authors intend to provide an outlook on prospective techniques, which are currently being researched. This review is intended for providing relevant clinical information to physicians. Therefore, several published studies between 2003 and 2021 were identified using PubMed, Google Scholar, or Research Gate and selected to be discussed for this purpose.

The following keywords were used: blood parameter measuring, invasive, non-invasive, in vivo blood components, blood gas monitoring, NIRS, pulse oximetry, capnometry, microdialysis, ICP measuring, (Continuous) Glucose Monitoring, Paratrend monitor, Marquet Cardiohelp, and radial catheter. Literature was searched for methods to measure blood gases (O_2_, CO_2_), pH, glucose, and lactate. Figure 1 illustrates the respective literature search methodology. There are two broad classes of methods for performing these measurements—either invasive (invasive in a medical context involves the introduction of instruments or other objects into the body or body cavities) or non-invasive. 

## 3. Non-Invasive Methods

In this section, we provide a detailed description of the most important and clinically used devices while also including a critical discussion regarding their benefit for the ICU and operation room (OR) with special emphasis on transcutaneous approaches.

### 3.1. Pulse Oximetry

Conventional pulse oximeter, which measures oxygen saturation in particular (and commonly also heart rate) [12] is the most used clinical device for a rapid survey of critical respiratory failure with limited resources and informative value. Pulse oximetry is a transdermal measuring method relying on spectrophotometry. The change in absorption properties of haemoglobin is correlated with the specific oxygenation level and used to register current oxygen saturation. Relative changes in light absorption are typically detected at two wavelengths (red at 660 nm and near infrared at 940 nm) used for the detection [13]. Pulse oximetry method offers several advantages such as the high measuring rate, easy application by placing the sensor on the patient’s finger, nose or even the ear, and a rather low-cost medical device. Nevertheless, there are some disadvantages and limitations. Pulse oximetry is not reliable in cases of decreased peripheral perfusion due to centralization or shock. In case of carbon monoxide intoxication, the measuring systems detect inadequately high oxygenation levels leading to a misinterpretation of the severe oxygenation deficiency. In anaemic situations, the blood oxygen level is reduced although the arterial oxygen saturation is normal. Therefore, an interpretation bias of peripheral capillary oxygenation might occur, which can result in the underestimation of respiratory insufficiency. In order to increase the reliability of the oxygen blood level, pulse oximetry is frequently combined with intermittent invasive arterial blood oxygen pressure analysis.

### 3.2. Capnometry and Capnography

Capnometry and capnography enable physicians to derive conclusions about ventilation as well as haemodynamics. Since the end-tidal carbon dioxide (etCO_2_) concentration represents the equilibrium between CO_2_ production (depending on metabolism) and elimination (depending mainly on ventilation and perfusion), changes can give important and quick information about the patient’s condition. For mechanical ventilation during surgical procedures, capnometry is used in addition to pulse oximetry in order to ensure adequate gas exchange by quantifying the end-tidal carbon dioxide (etCO_2_) concentration. Those results reflect alveolar pCO_2_-concentrations, which are close to arterial concentrations in patients with physiological perfusion and ventilation [14]. Overall, etCO_2_ is used as an adjuvant parameter guiding successful endotracheal intubation. Additionally, during cardiopulmonary reanimation (CPR), etCO_2_ is recommended as a validating tool for the efficiency of CPR as well as the possible entry of ROSC (Return of Spontaneous Circulation) [15]. The ERC-Guidelines 2021 [16] state that high and rising etCO_2_ levels are correlated with elevating survival rates. Furthermore, capnometry might help to detect early respiratory failure, for example during non-invasive ventilation or spontaneous breathing with intravenous sedation thus increasing patients’ safety [17]. Capnometry is cost-effective and facilitates almost continuous measuring. Nevertheless, one of its main disadvantages is the premise of a closed ventilation system [18]. In some cases, such as pulmonary embolism, shock, or chronic obstructive pulmonary disease (COPD), the arterial pCO_2_ may not correlate with the etCO_2_ results, which might limit the clinical value. The dead volume of the whole ventilation system is increased by 6 mL, which is mostly negligible in adults, but could be relevant in neonatal intensive care.

### 3.3. Transcutaneous Blood Gas Monitoring

Transcutaneous blood gas monitoring is frequently used in neonatal ICUs [19]. However, this method is not limited to neonates, but can also be applied to adults. Latest research focuses on a study “Transcutaneous Carbon Dioxide Monitoring in Electrophysiological Procedures in Deep Sedation (TRACES)” that will be completed by 04/2022 [20]. Another state-of-the-art technology enables the detection of tissue oxygenation. Andrews et al. summarized that TcPO_2_ (transcutaneous partial pressure of oxygen) measurements may provide a more significant prognostic value than the ankle-brachial indices relating to healing after partial foot amputation [21]. Transcutaneous blood gas monitoring depends on factors such as arterial blood pressure, skin blood circulation, skin respiration, and diffusion conditions [22]. The sensors, which measure transcutaneous blood gases, are placed on the patient’s skin. A heating element warms the skin and increases the perfusion of the tissue. Currently, receivable devices need to be calibrated every 12 h using a rather complex procedure. Several gas measurement techniques have been enhanced from electrochemical [23] to optical fluorescence quenching oxygen measurements [24]. Developing a miniaturized photoacoustic gas sensor with long calibration intervals intended as an upgrade for the conventional electrochemical sensors has been proposed recently as an alternative [25].

For surgical procedures with high risk of reduced cerebral perfusion or thromboembolic occlusion, standard cardiovascular monitoring, e.g., SpO_2_ (peripheral oxygen saturation) and blood pressure might be insufficient for detecting brain tissue hypoxia. 

### 3.4. Near Infrared Spectroscopy (NIRS)

Near infrared spectroscopy (NIRS) offers non-invasive monitoring of tissue oxygenation [26]. Besides the application during surgical procedures, NIRS is used for patients with traumatic brain injury (TBI) such as intracranial hematomas [27]. NIRS is a reliable and approved device, which has various clinical scopes of application. Furthermore, this method could facilitate identifying patients with critical perfusion/oxygenation mismatch in preclinical settings [28]. Specific probes called optical terminals or optodes, are placed on the scalp skin [29]. As all molecular substances of the human blood exhibit characteristic bands in the infrared and near-infrared spectra within the range from 630 to 1350 nm, optical detection is feasible. Light beams penetrating the skin are mainly absorbed by oxyhaemoglobin and deoxyhaemoglobin. By using different signals detected by the spectrometer, the concentration of saturated haemoglobin can be calculated [30]. Moreover, this application could prevent an underestimation of tissue hypoxia in cases of centralization and shock when other measuring systems come up against their technical limits. In these situations, adequate cerebral oxygenation and cranial perfusion are difficult to maintain. Optodes for NIRS are expensive and only for single use. Therefore, they are not applicable in long-term monitoring. Furthermore, it is necessary to know that NIRS should only be interpreted as a trend monitor. It is required to measure a baseline at the start and evaluate the changes or drop in rSO_2_ (regional oxygen saturation). Additionally, there are still discussions about the appropriate thresholds to start an intervention [31]. As NIRS is a transdermal non-invasive detection system like pulse oximetry, different factors such as skin pigmentation or myoglobin concentration might influence the measurement results. During liver surgery, especially liver transplantation, it is necessary to keep the current brain oxygenation under surveillance to avoid neurological failures due to hepatic encephalopathy [32]. In those surgery scenarios NIRS is increasingly applied for monitoring the current brain oxygenation state. Next to methaemoglobin or carboxyhaemoglobin potentially influencing the results, bilirubin and biliverdin also can lead to significantly aberrant low values [33]. The higher the presurgical bilirubin values, according to laboratory examination right before intervention, the lower the measured oxygen levels offered by NIRS. 

### 3.5. Transcutaneous Bilirubinometry

Due to higher bilirubin levels in newborns, some devices exist for measuring bilirubin by transcutaneous bilirubinometry, which offers a non-invasive bilirubin assay [34,35]. Transcutaneous bilirubinometry is suitable as a screening tool considering the start of phototherapy for babies with hyperbilirubinemia. Systems are working in the visible light within the spectral range between 380 and 760 nm, which is reflected by the skin, and mathematically subtracts the absorption contributions of interfering factors such as haemoglobin or melanin. The results offer bilirubin concentration in the capillary beds and subcutaneous tissue [36].

### 3.6. Non-Invasive Lactate Sensors

In the past several years there has been a development of non-invasive lactate sensors which are able to measure lactate levels in humans’ perspiration continuously [37]. Besides electrolyte concentrations like chloride or sodium, it is also possible to measure unphysiological lactate levels which indicate the metabolic pathways in context of basal or physical energy rate. The cited publication states the correlation between blood and sweat lactate levels as the principle for the depicted measuring system [38]. In most cases, the measurement is based on electrochemical sensors also as integrated systems enabling the detection of different analytes simultaneously [39].

### 3.7. Sweat Sensors

Sweat sensors have already been available in patient care for several years; electrochemical lactate sensors dominate the market because of their sensitivity and low cost [40]. Sweat is easily accessible through non-invasive measuring systems, which represents one of the main advantages. Furthermore, sweat sensors might also wirelessly transmit the data about measured signals, which facilitates their application on critically ill patients in particular. Nevertheless, the reported sensors always need physical contact for continuous sampling and measurement of lactate molecules. In addition, movements of the patient might lead to lapses in the results. There is still some potential for further development, specifically considering the integration of multiple analyte measurements within one sensing tool. Development of wearable sweat sensing platforms which can detect multiple analytes is already an active research topic with promising results [41].

In summary, currently available non-invasive devices for clinical use are mainly based on transdermal approaches such as infrared spectroscopy and other optical as well as chemical measuring methods. Additionally, infrared spectroscopy detects amongst other components for example blood glucose and urea. The measured IR spectra, also called the IR fingerprint absorptions, require chemometric processing through Fourier transform [42] or with quantum cascade lasers (QCL). Table 1 summarizes the main devices and approaches in clinical use for non-invasive measuring. 

## 4. Invasive Methods

For critically ill patients in the ICU, decisions based on specific and reliable measurements of crucial blood parameters might improve clinical care. For invasive measurements, transcutaneous or surgical placement of catheters or sensors is required. These interventions might lead to severe complications, e.g., bleeding, organ damage, or infections. 

### 4.1. Oxygen Sensors

A novel fluorescence quenching technology with an uncoated probe tip introduced by OceanOptics, Inc. (Orlando, FL, USA) can measure pO_2_ in blood and brain tissue [43]. A ruthenium complex at the sensor emits light at 600 nm upon excitation at 475 nm, and molecular oxygen in the bloodstream quenches this fluorescence. This principle allows continuous as well as highly reliable measurements. The probe is nearly immune to environmental changes concerning for example pH or ionic strength. Animal experiments, performed in 2011, proved this technique to be a reliable alternative for fast measuring of oxygen pressure levels in blood and brain tissue [44]. The Foxy-pO_2_ (OceanOptics, Dunedin, FL, USA) results corresponded with conventional Clark-type electrode measurement results as the study states. Besides that, both macrocirculation, tested in the thoracic aorta, and microcirculation of brain tissue can be measured accurately. Nevertheless, the probe is only usable for the detection of pO_2_; it does not combine the detection of different parameters, which impedes its wide usability.

### 4.2. Intracranial Probes

In addition to the detection of specific blood parameters in brain tissue non-invasively through NIRS, invasive measuring is available for a wide spectrum of parameters. Intracerebral pressure probes are mostly used within neuro-intensive care. Several patients must be monitored in case of traumatic brain injury (TBI), intracerebral bleeding, or central nervous system infections. Individuals with TBI often benefit because the average intracranial pressure (ICP) in the first 48 h after the triggering occurrence seems to predict mortality and functional outcome after 6 months independently. Due to impaired consciousness caused by other systemically effective diseases such as acute liver failure, end-stage kidney failure or hypertensive encephalopathy, as well as different congenital diseases such as hydrocephalus at pediatric units, the use of intracranial probes might also be required [45]. Besides the “standard” monitoring of ICP, some devices were developed for measuring specific tissue alterations. One type of these devices is a specific microcatheter, inserted into the brain tissue. Measuring is performed by an optode, a specific oxygen microsensor, based on the oxygen-quenching of fluorescence light [46,47]. The offered results include ICP, pO_2_, pH levels, lactate, and temperature [48]. In combination with the microdialysis method described below, it is also possible to monitor cerebral metabolism. This device has already been available in clinical care for several years. Although it can only be used for a narrow range of patients, the application might enhance patient care [45]. Intracranial probes offer exact measuring results which enable fast recognition of changing parameters. Nevertheless, there are various complications possible such as infections, further increase of intracranial pressure through the application of the probe, and intracranial bleedings. 

### 4.3. Microdialysis

Microdialysis is a minimally invasive and rather diverse sampling method already available for patients in neurosurgical intensive care. It can monitor continuously the concentrations of free, unbound analytes such as neurotransmitters and energy substrates in the extracellular environment, e.g., glucose, lactate, cyclic AMP, or histamine. Besides the electrolytes, the monitoring includes acetylcholine and GABA, ethanol and cortisol levels, as well as other hormones such as follicle-stimulating hormone [49]. Moreover, it is possible to analyse exogenous drugs including antidepressants and antipsychotics with pharmacological effects on the brain [50]. Technically, a probe is inserted into an external medium, which is in most cases a special tissue such as brain tissue or blood [51]. This probe consists of a semipermeable membrane that permits free passive transport of selected solutes moving along their concentration gradient. The measuring works through flow rate and dynamic no-net-flux or zero-net-flux methods [52]. Microdialysis offers a wide range of results, which is one of its main advantages. Furthermore, microdialysis probes are smaller than ones provided by other devices, which affects the surrounding tissue much more gently. Nevertheless, the different measured transmitters and ions might have an impact on the detection of other analytes as well as the baseline. This makes the interpretation of the results more difficult and reduces precision below the required level [53]. 

### 4.4. Paratrend Monitor System

Another important device is the Paratrend monitor. It is widly used for critically ill patients in the ICU and during different types of surgery such as open thoracic or great vessel surgery [54]. While using this device, different arterial blood gases, e.g., pO_2_ and pCO_2_, are continuously measured directly in the vessel [55]. It is possible to monitor pH and temperature as well. The measuring access is provided through an invasive arterial catheter which is normally inserted into the radial or femoral artery. Paratrend uses a fluorescence-based method [56]. A semipermeable membrane guarantees gas and water exchange. The dissoluble components can be measured using light at wavelengths of 812 nm as part of the near infrared spectrum and 660 nm as visible red light [57]. Moreover, pCO_2_ and pH values are provided by luminescence methods, while a specific temperature electrode and a Clark electrode provide temperature and pO_2_ level results, respectively. The Paratrend monitor system, distributed by Diametrics, offers continuous measuring with high accuracy especially for pH detection [58]. Because of the high number of electrodes, a whole monitoring system is needed for every single patient, which requires technical and personnel expenditure. Although there are many advantages of the Paratrend monitor, the original device has been on the market only up to 2003. At this point, the Paratrend 7 system as well as Neotrend have been the only commercially available multiparameter continuous intravascular blood gas monitoring systems. Clinical observations demonstrated that the performance of Paratrend in the clinical setting was not satisfying compared to the performance reported under laboratory settings [56]. PO_2_ for example was affected by a bias especially in the higher ranges. The impact on patients outcome has not been examined in a wide setting [56].

Recently, a novel intravascular sensor was developed by Oxford Optronics. Construction of this sensor is based on the precedent Paratrend system. It has already been tested on animals with promising results. Up to this point, it is only intended for laboratory, industrial, and research use and has not been used in patients yet [59]. 

### 4.5. Devices in Combination with ECMO Systems

Certain devices are available that can measure different blood components in vivo in combination with the ECMO system such as the “Marquet Cardiohelp” module [60]. Indications for ECMO are for example ARDS, aortic surgery, or extracorporeal CPR (eCPR). Furthermore, it can be brought in for rewarming of a patient due to hypothermia. Blood flows from the vessel through the cannula into the device outside the human body and comes into contact with different combined sensors and probes such as a venous probe, external temperature sensor, and pressure sensor [61]. These sensing components are developed with variants of measuring systems including optical or electrochemical methods. Because of this multiplicity, many values are being measured simultaneously including temperature, arterial blood pressure, and Hb/HCT. One main benefit is that Marquet Cardiohelp can be used for a time period of up to thirty days. Besides the ECMO cannulas, no further invasive catheter has to be inserted. Because of its flexibility in usage, the device might be brought into transportation situations [62].

### 4.6. Continuous Measurement of pH, pCO_2_, pO_2_

Another example of a device under development is a catheter for continuous intra-arterial blood gas monitoring (CABG) [63]. The TrendCare System, invented by Diametrics Medical, consists of a shorter catheter body with a measuring tip. The sensor itself consists of three optical fibres measuring pH, pO_2_ and pCO_2_ as well as a thermocouple for temperature measuring. The pO_2_ measuring is based on a pure optode system which replaced the Clark electrode used in the preceding conventional Paratrend system. A prospective study with 25 adult patients in ICUs who required mechanical ventilation has already been performed by testing the catheter for a period of four days in total [64]. The results show an adequate performance compared to the conventional blood gas analysis (ABL 610). Nevertheless, the clinical use has been accompanied by different problems related to the intravascular application, so the system has not been widely used on patients yet. The system and testing method have been a model for further developments integrating intravascular blood gas analysis. Promising in-vitro testing has already been performed as well as animal experiments with a further established radial catheter. The catheter consists of a fluorescence-based optical measuring system with three optical chambers in progress to detect pH levels as well as pCO_2_, pO_2,_ and temperature in particular [65]. 

### 4.7. Continuous Glucose Monitoring

Continuous glucose monitoring (CGM) is mainly used by patients suffering from diabetes. Meanwhile, there are different commercially available systems. 

Flash glucose monitoring systems are among the most recent CGM technologies [66]. A sensor with a small needle is placed on the patient’s arm. CGM methods are based on measuring the interstitial blood glucose level subcutaneously. Several studies have shown that the rate of diabetes-related hospitalization and hypoglycemia decreased by using CGM [67,68,69]. However, it should be noted that the accurate measurement with this technology has a time lag compared to conventional blood glucose monitoring.

It has been criticized that the optimal blood glucose target remains unclear in critically ill patients [70]. The NICE-SUGAR Study concludes that intensive glucose control might lead to moderate and severe hypoglycemia, both of which are associated with an increased risk of death [71].

The COVID-19 pandemic has accelerated the trend of collecting data remotely from patients so the devices are increasingly included in the hospital setting [72].

The explanations show various devices intended for different clinical situations. Most of them are in daily clinical use even though some show a narrow range of indications and patients they might be used for. The specificity of a few devices, for example such as pulse oximeters, is questionable. Because of these reasons, significant further development is necessary, not only regarding the improvement of current devices but also the measuring approaches. Table 2 summarizes the main devices and approaches in clinical use for invasive measuring.

## 5. Towards the New Generation of Biosensing Systems

### 5.1. Point-of-Care-Devices

Point-of-care devices are available at the bedside for certain parameters, e.g., blood gas analysis and coagulation tests, but some samples still must be analyzed in central laboratory units. Even if point-of-care devices and rapid processing central laboratory services are available, these measurements remain discontinuous and do not enable individualized treatment approaches in dynamic and rapidly changing critical situations. Real- time measurement of key parameters (e.g., lactate, electrolytes, glucose, oxygenation, and acid-base status) that are currently only available as a separate blood test, would improve patient care and individualized treatment. Smart electrochemical micro-sensors can provide continuous monitoring of essential parameters, without increased consumption of blood or use of additional sensor interfaces. Such sensors might integrate all necessary components on a smart catheter for reagent-less and real-time analysis, including selective membranes that protect the surface from deterioration. Haemocompatible coating prevents blood clotting and inflammation at the sensor surface. In the future, this approach might enable patient centered individualized therapy guidance and therefore improve quality of care, patient safety, and outcome in critically ill patients. In this section, we review the most promising strategies which can contribute to the area of bedside patient monitoring based on the use of electrochemical sensors.

Many of the new strategies still rely on optical transduction modes which have shown great potential but still require large setups with limited options for multiplexing by integrating small arrays. These approaches can make use of implantable fibres modified with responsive fluorescent hydrogels, as hydrogels are known to allow for long-term implantation. Heo et al. introduced polyethylene glycol-bonded polyacrylamide hydrogel fibres in the ears of mice which could glow at fluorescence intensity correlated with blood glucose level for up to 140 days (Figure 2a) [73]. Sawayama et al. developed a subcutaneous device which included a boronic acid-based glucose responsive fluorescent dye protected in a hydrogel doped with antioxidant enzymes (superoxide dismutase and catalase), thus avoiding the degradation of the dye by hydrogen peroxide (Figure 2b) [74]. The implantable unit could operate for 28 days in rats.

### 5.2. Miniaturization Processes and Biosensor Development

A transition to purely electrical or electrochemical readout mode can help to achieve a smaller sensor size, enabling the integration of further sensors in the same area and additional circuitry for wireless connectivity, and also avoiding complex or large equipment for signal measurement. In an electrochemical sensor for example, redox reactions happening at the surface of a working electrode contribute to variations in the measured electrical signal, which can be specific toward the analyte of interest if specific receptors are incorporated. The desire to achieve electrical and electrochemical measurements of blood parameters can be traced back to the 1950s and 1960s, when L.C. Clark proposed the use of polarography [75] to measure oxygen tensions and the modification of electrodes with glucose oxidase to monitor glucose levels due to its inherent oxygen dependency [76]. With the latter, the first biosensor ever was born. However, blood withdrawal was required for the continuous monitoring. During the past decades, the synergy between micro- and nano-fabrication techniques, chemistry, and biology resulted in the development of ultrasensitive and ultraminiaturized devices for a vast range of biomarkers and parameters [77,78,79,80]. Unfortunately, when it comes to in vivo usage, the situation is highly challenging for a successful result, due to the many barriers that living organisms pose to the biosensor function such as biofouling, inflammation, fibrosis, release of lysosomal contents, blood coagulation, and thrombus formation [81,82]. Some strategies can be found in the scientific literature to tackle the problem of sensor signal drift. On one hand, the system can incorporate reference redox reporters or reference sensors, serving as a drift-correcting reference [83,84]. Figure 3a shows an electrode surface modified by a specific aptamer for cocaine detection in blood. The aptamer contains two redox reporters, both with different redox potentials, avoiding overlapping signals. The first one is independent of the cocaine presence, while the second one approaches the electrode surface when the aptamer folds catching the cocaine molecule. Samples with cocaine therefore increase the signal at the redox potential of the second reporter, while the first reporter signal remains stable. Any drift unrelated to the cocaine concentration should affect both reporters, allowing the correction of the signal by a simple signal subtraction. However, ideally the surface should be protected to avoid deterioration of both reference and specific sensor signals. Here, the hydrogels previously mentioned in optical sensors can play an important role in protecting the electrode surface from deterioration and filtering the approaching excess of biological material. The same working group that demonstrated the cocaine sensor also published a preliminary work (preprint) [85] where the aptamer-modified surface (for kanamycin antibiotic in this case) was modified with an agarose hydrogel, achieving a good baseline stability in veins, muscles, bladder, or tumors of living rats without the need for the drift correction reporter (Figure 3b). Others opted for electrospun membranes based on polyurethane and gelatin [86] or polyurethane and nafion [87] to protect glucose biosensors. 

Finally, another alternative could be the use of external, minimally invasive devices attached to the skin (wearable sensors) which make use of reverse iontophoresis to refiltrate intravascular molecules toward the cathode electrode, thus measuring blood molecules but not in situ. Chen et al. for example fabricated a flexible paper battery on skin-like glucose biosensors, where intravascular blood glucose was refiltrated to the interstitial fluid and later towards the skin, helping to obtain a good correlation between the measured signal and that obtained via fingerpricking using a commercial glucometer (Figure 3c) [88].

In all cases, the biological receptor molecules such as the enzymes used for glucose detection can be substituted with purely inorganic alloys or composites which would solve stability or functionality issues arising from biomolecule conformational changes. Examples can be found using Au-Ni alloys [89], hollow nanoporous CuO/ZnO microstructures [90], or CuO Modified Vertically-Grown ZnO Nanorods [91], all of them showing catalytic activity towards glucose. However, many of the nonenzymatic sensors require special conditions, such as e.g., alkaline pH in case of CuO-based glucose sensors, limiting the application to in vitro blood analysis.

## 6. Complete Electronic Sensing Solutions

To provide the multiparametric analysis of the vital signs, e.g., in blood, the sensor platform has to be embedded into the proper conditioning electronics system. This involves processing strategies to achieve multiplexing capability and noise or interference reduction, as well as strategies for the information transfer. Some of the possible solutions are described below, taking glucose as the representative analyte.

Currently available off-the-shelf glucose monitoring devices suffer from a number of practical issues, for example, the hygiene of the measurement area affects measured blood glucose levels, risk of infections, tissue deformation, etc. Such issues indicate that there is a need for further technological developments in order to provide affordable, reliable, and real-time measurements without any patient discomfort. The radio frequency (RF)/microwave sensing of blood glucose levels is a technique in the focus of current research efforts. Furthermore, the progress in the field of wireless technology experienced in the past decades has awaken widespread interest in the study of the influence of electromagnetic waves on biological tissues [92,93].

Next, Omer et al. demonstrated a wearable sensor comprising hexagonal-shaped complementary split ring resonators (CSRR) operating in the industrial, scientific, and medical band (ISM) 2.4–2.5 GHz for the detection of blood glucose levels [94]. Moreover, they investigated the electromagnetic properties of solutions with different glucose concentrations in the 300 MHz–67 GHz frequency band, and proposed the use of a mono-static multi-channel frequency modulated carrier wave (FMCW) radar operating in the 57–64 GHz frequency range to perform the detection [95]. Saha et al. presented a sensing system that uses two patch antennas operating at 60 GHz and used the permittivity of the samples to calculate the different levels of blood glucose [96]. 

## 7. Conclusions

Real-time blood parameter monitoring devices for patients in the ICU have been reviewed in this article. Together with these devices, invasive and non-invasive techniques for the measurement of various blood parameters relying on different working principles have also been discussed. There are still multiple opportunities to improve the performance of existing devices and techniques as well as the possibility to seek innovative approaches for improving the quality of care. 

## Figures and Tables

**Figure 1 jcm-11-02408-f001:**
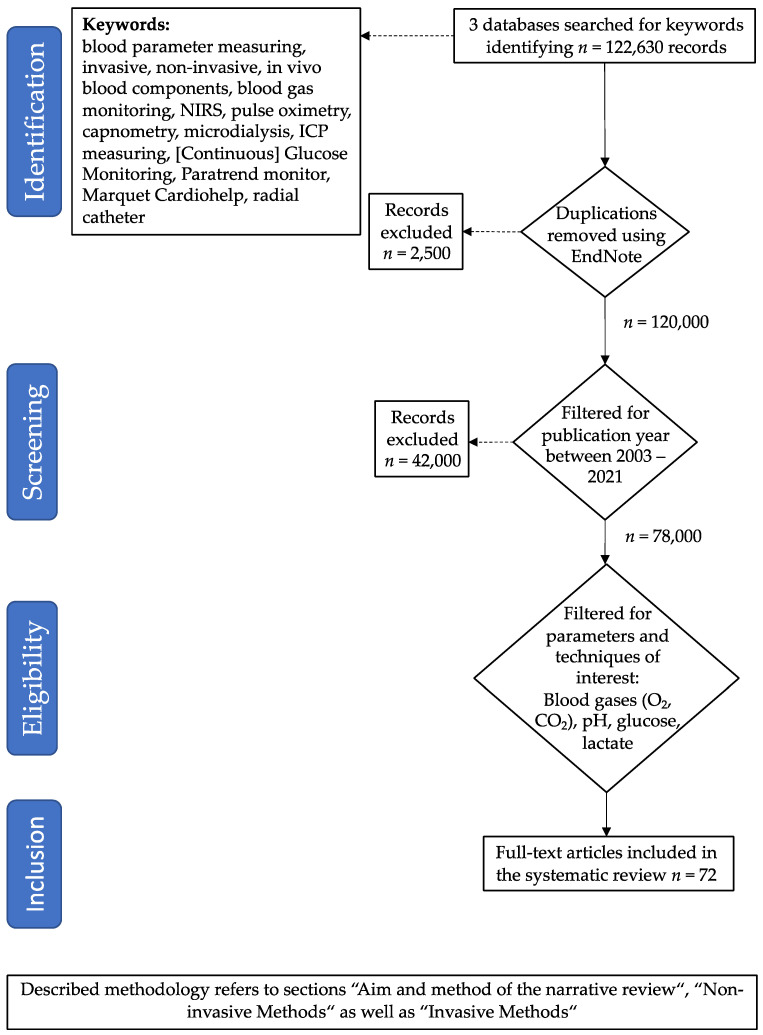
Literature search method used in this review article.

**Figure 2 jcm-11-02408-f002:**
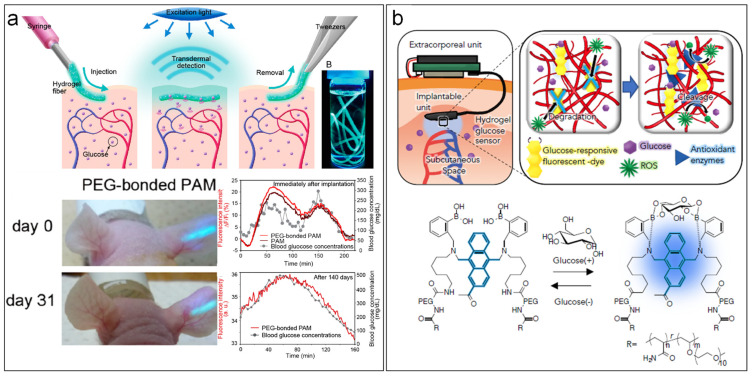
Implantable devices for blood-glucose monitoring. (**a**) Hydrogel-modified fibre with fluorescence responsivity operating in the ears of mice and responding to blood-glucose levels. Adapted with permission from Yun Jung Heo et al. [73], (**b**) Subcutaneous device with boronic acid-based glucose responsive fluorescent dye protected within a hydrogel doped by antioxidant enzymes. Adapted with permission from Sawayama et al. [74].

**Figure 3 jcm-11-02408-f003:**
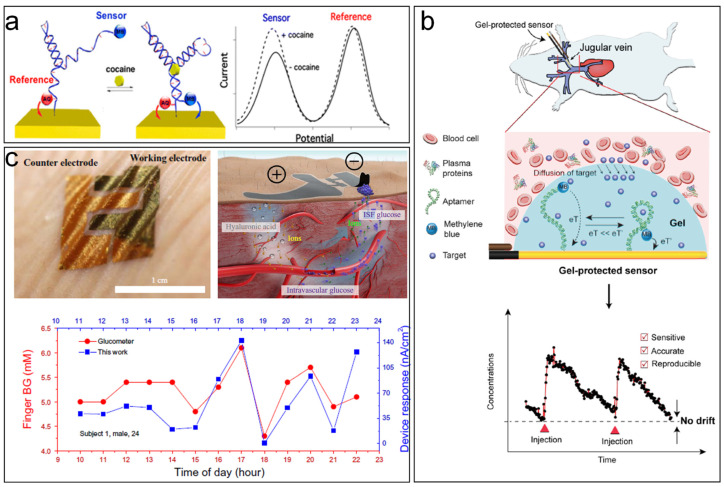
Electrochemical blood parameter measurement techniques. (**a**) Cocaine detection using aptamer-based electrochemical sensor with a reference redox-reporter for signal drift correction. Adapted with permission from Li et al. [83], (**b**) In vivo kanamycin measurement with aptamer-based electrochemical sensor protected with agarose hydrogel. Adapted with permission from Li et al. [85], (**c**) Skin-like electrodes for the measurement of blood glucose by refiltration to the interstitial fluid. Adapted with permission from Chen et al. [88].

**Table 1 jcm-11-02408-t001:** Non-invasive methods.

Measuring Method	Area of Detection	Measured Parameters	Scope of Application (Selection)	References
**Pulse oximetry**	Fingertip, earlap, tip of the nose	Oxygen saturation	Evaluation of vital parameters in various settings, esp. respiratory failure	Hafen et al., 2021, ref. [12] Chan et al., 2013, ref. [13]
**Capnometry**	Endotracheal ventilation tube	Exspiratory pCO_2_	Efficiency of CPR, monitoring of intubation, detection of respiratory failure	Nassar et al., 2016, ref. [17] Kupnik et al., 2007, ref. [18]
**NIRS (Near Infra-Red Spectroscopy)**	Scalp	Brain tissue oxygenation	General surgery, traumatic brain injury (TBI)	Sen et al., 2016, ref. [27] Scheeren et al., 2012, ref. [28]
**Bilirubinometry**	Skin, mostly forehead/sternum	Subcutaneous bilirubin concentration	Consideration of therapy, continual monitoring	De Luca et al., 2008, ref. [34] Rizvi et al., 2019, ref. [35]

**Table 2 jcm-11-02408-t002:** Invasive methods.

Measuring Method	Area of Detection	Measured Parameters	Scope of Application (Selection)	References
**Intracranial probes**	Intracranial (brain tissue)	ICP, pH, pO_2_, lactate, temperature	traumatic brain injury, intracerebral bleeding, infection	Evensen et al., 2020, ref. [45] Keller et al., 2011, ref. [47] Seule et al., 2016, ref. [48]
**Microdialysis**	Intracranial (in the blood cycle)	Neurotransmitters, metabolic substrates	Neurosurgical and general intensive care	Chefer et al., 2009, ref. [49] Hammarlund-Udenaes et al., 2017, ref. [52] Shippenberg et al., 1997, ref. [53]
**Paratrend monitor**	Blood vessels, mostly radial or femoral artery	pH, pO_2_, pCO_2_, temperature	ICU, thoracic and great vessel surgery	Ganter et al., 2003, ref. [56] Mantouvalou et al., 2004, ref. [57]
**Marquet Cardiohelp (ECMO)**	Sensors and probes within the device	Blood pressure, temperature, hemoglobin, hematocrit	ARDS, aortic surgery, extracorporeal CPR, rewarming	Staudinger et al., 2016, ref. [60] Alwardt et al., 2015, ref. [61] Arlt et al., 2011, ref. [62]
**CGM (Continuous glucose monitoring)**	Subcutaneous, mostly arm	Interstitial blood glucose level	Diabetes monitoring	Bailey et al., 2021, ref. [66] Krinsley et al., 2017, ref. [70] Perez-Guzman et al., 2021, ref. [72]

## Data Availability

Not applicable.

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
