# Peer review of "Real-Time Monitoring of Blood Parameters in the Intensive Care Unit: State-of-the-Art and Perspectives"

_jcm, 2022, doi:10.3390/jcm11092408_

Round 1

Reviewer 1 Report

The authors summarized the current available non-invasive and invasive measuring methods in the ICU.

1. The epidemiological description of the disease in the introduction is too much, since this article is mainly to review the parameters and instruments of monitoring, it is recommended to shorten this section.

2. Why Invasive Ventilators are not concluded in invasive-methods? 

3. Some references are too old and should be updated.

Author Response

  1. The epidemiological description of the disease in the introduction is too much, since this article is mainly to review the parameters and instruments of monitoring, it is recommended to shorten this section.

Response:

We shortened the introduction according to your suggestion.

2. Why Invasive Ventilators are not concluded in invasive-methods? 

Response:

Thank you for this valid point. We decided to put capnometry/capnography in the non-invasive section since the method per se is non-invasive and could be used with both invasive and non-invasive ventilation.

3. Some references are too old and should be updated.

Response:

We re-checked the references, however, some of the hallmark papers in this field are rather old, in figure 1 we illustrated our search methodology to demonstrate that we were not neglecting more recent publications. Especially the outlook on new developments in this field contain more recent publications from the last two years.

Reviewer 2 Report

Dear Authors:

Thank you for the opportunity to review your manuscript.

Author Response

  1. The methods section does not sufficiently explain the methodology used for the systematic review. There was no information on the selection criteria or process used for article selection. Furthermore, I recommend that you include a flowchart of the literature review including the keywords used for retrieval and the number of relevant documents.

Response:

Thank you for this valuable comment. We included Figure 1 in the resubmission to illustrate our search methodology as suggested.

2. In the resultssection (P.3, Line 98–P11, Line 485), addition of subheadings such as Pulse oximetry, Capnometry, etc., would improve clarity and facilitate better understanding of the findings.

Response:

We inserted the subheadings according to your suggestion in the revised manuscript.

3. In Tables 1 and 2, I recommend that you show the references and their dates of publication of facilitate an understanding when each real-monitoring tool was discussed

Response:

We modified Table 1 and 2 in the resubmitted manuscript according to your suggestions and inserted the respective references in the last column.

Round 2

Reviewer 1 Report

Figure 1 should be reuploaded.

Author Response

We replaced Figure 1 accordingly.